# Double Electromagnetically Induced Transparency and Its Slow Light Application Based on a Guided-Mode Resonance Grating Cascade Structure

**DOI:** 10.3390/ma13173710

**Published:** 2020-08-21

**Authors:** Guofeng Li, Junbo Yang, Zhaojian Zhang, Yuyu Tao, Lingjun Zhou, Huimin Huang, Zhenrong Zhang, Yunxin Han

**Affiliations:** 1Guangxi Key Laboratory of Multimedia Communications and Network Technology, School of Computer, Electronics and Information, Guangxi University, Nanning 530004, China; 1813391006@st.gxu.edu.cn (G.L.); taoyuyu@st.gxu.edu.cn (Y.T.); 1813391017@st.gxu.edu.cn (L.Z.); 1813301006@st.gxu.edu.cn (H.H.); 2Center of Material Science, National University of Defense Technology, Changsha 410073, China; yangjunbo@nudt.edu.cn (J.Y.); 376824388@alumni.sjtu.edu.cn (Z.Z.)

**Keywords:** guided-mode resonance, electromagnetically induced transparency, waveguide grating structure, slow light

## Abstract

In recent years, the achievement of the electromagnetically induced transparency (EIT) effect based on the guided-mode resonance (GMR) effect has attracted extensive attention. However, few works have achieved a double EIT-like effect using this method. In this paper, we numerically achieve a double EIT-like effect in a GMR system with a three-layer silicon nitride waveguide grating structure (WGS), using the multi-level atomic system model for theoretical explanation. In terms of slow light performance, the corresponding two delay times reach 22.59 ps and 8.43 ps, respectively. We also investigate the influence of wavelength detuning of different GMR modes on the transparent window and slow light performance. Furthermore, a wide-band flat-top transparent window was also achieved by appropriately adjusting the wavelength detuning between GMR modes. These results indicate that the EIT-like effect in the WGS has potential application prospects in low-loss slow optical devices, optical sensing, and optical communications.

## 1. Introduction

The electromagnetically induced transparency (EIT) effect is caused by quantum interference of different excitation paths in a three-level system; it is also one of the important technologies to achieve the slow light effect [1,2]. The EIT effect happens where the pump light passes through the medium without absorption under the action of strong resonance coupling light. Similar to EIT is extraordinary light transmission (ELT), which can add an external magnetic field to make light pass through the opaque film with high transmission [3]. The transparent window of EIT is usually accompanied by extremely strong dispersion [4].

Over the past few decades, the EIT effect has aroused the interest of many researchers due to its unique ability to reduce and control the speed of light [5,6,7]. In addition, EIT has many potential applications in sensing [7], nonlinear optics [8], and cavity quantum electrodynamics [9]. Researchers have to resort to classic optical systems for an EIT-like effect, since EIT effects in atomic energy systems happen under strict experimental conditions, such as ultra-low temperature, hence limiting their practical application. Shanhui Fan successfully accomplished the EIT-like effect in an optical microcavity system in 2002 [10], and there has been increasing work using this approach to implement the EIT-like effect [11,12]. Since then, EIT-like effects have also been achieved in plasmonic systems [13,14] and metamaterial structures [15,16]. Sun-Goo Lee developed the EIT-like effect based on the guided-mode resonance (GMR) effect in a waveguide grating structure (WGS), which contains two planar dielectric waveguides and a sub-wavelength grating [17], where the transparent window appears in the transmission dip by coupling high-Q and low-Q resonance modes in different dielectric waveguides.

In recent years, there has been much research interest in obtaining EIT-like effects by utilizing the GMR effect. Sun-Goo Lee presented two photonic systems that make it possible to achieve the GMR-based polarization-independent EIT-like effect, with both systems including two planar dielectric waveguides and a two-dimensional photonic crystal [18]. Yaru Sun reported a new planar metamaterial composed of a two-ring-resonator unit cell based on the GMR effect to achieve an EIT-like effect [19]. Han achieved an EIT-like effect with a Q value of 288,892 in a GMR system that includes two silicon grating waveguide layers stacked on CaF_2_ substrates [20]. Previously, we reported a double EIT-like effect with two high Q values in a GMR system, where both waveguide layers have two dielectric gratings in one period, and two narrow-band transparent windows appear in the two transmission dips on account of the coupling of different GMR modes [21]. However, few works have achieved a double EIT-like effect by taking advantage of the GMR effect in a WGS. In recent years, cascade resonant grating has attracted extensive attention and research due to its versatility. Not only can it be used to design filters [22,23,24], but optical phenomena very similar to EIT characteristic spectra were also observed in some work [24,25,26]. Therefore, it will be an interesting topic to study the EIT-like effect combining the characteristics of the GMR effect and cascade resonant grating.

In this paper, a double EIT-like effect is investigated numerically based on a GMR system including three silicon nitride (Si_3_N_4_) grating waveguide layers (GWLs) stacked on SiO_2_ substrates. Compared with our previous research [21], this work makes two transparent windows appear simultaneously in a transmission dip. The corresponding optical phenomenon can be explained by the multi-level atomic energy system model. We have discussed the dual-channel slow light performance of the system. We also present the effect of wavelength detuning of different GMR modes on the position of the transparent window and slow light performance by changing the grating width. Moreover, a broadband flat-top transparent window can be obtained through an appropriate detuning of the response wavelength between GMR modes. This work paves the way for high-performance slow light devices and has potential applications in optical communications.

## 2. Materials and Methods

The optical characteristics of the GMR system were investigated by the three-dimensional (3D) finite-difference time-domain (FDTD) method. The mesh accuracy was set to *λ*/26 to maintain the convergence of all simulation results. The schematic diagram of the GMR system with triple cascaded resonant gratings is shown in Figure 1a. The “Boundary conditions” were “Periodic” in the X and Y directions, and “perfectly matched layer (PML)” in the Z direction. The mesh accuracy in the three GWLs were Δx = 10 nm and Δz = 20 nm. The refractive index data of SiO_2_ [27] and Si_3_N_4_ [28] were derived from the original publication. The background refractive index of the system was *n_s_* = 1.

The light entered the system vertically and was a plane wave polarized in the X direction (TM polarization) in the near-infrared band. The period *P* of the grating was 809 nm. The thickness *t* of the Si_3_N_4_ waveguide was set to 900 nm. The three gratings (G_1_, G_2_, G_3_) had a depth *D* of 200 nm and a width of 400 nm (i.e., *W*_1_
*= W*_2_
*= W*_3_). In other words, the parameters of the three GWLs were exactly consistent. A SiO_2_ layer with a thickness *d*_1_ of 630 nm was placed on the top layer of the structure in order to reduce the reflected light. The distance *d*_2_ between GWL_1_ and GWL_2_ was 690 nm; the distance *d*_3_ between GWL_2_ and GWL_3_ was also 690 nm.

## 3. Results

We firstly calculated the transmission spectrum when there was only a single GWL in the GMR system. The corresponding transmission spectra are shown in Figure 1b. The solid lines of cyan, blue, and green are the transmission spectra of GWL_1_, GWL_2_, and GWL_3_, respectively. They all had a transmission dip with a center wavelength of 1550.44 nm (*λ_GMR_*_1_ = *λ_GMR_*_2_ = *λ_GMR_*_3_), which exhibits GMR characteristics of wide resonance and high reflection. However, their sideband transmittances were slightly different because their optical properties are easily affected by the surrounding medium. The top SiO_2_ layer had an effect on the resonance wavelengths of guide modes. To this end, the thickness of the SiO_2_ coating layer was adjusted to the appropriate parameters not only to ensure that the three GWLs had the same resonance wavelength, but also to maximize the transmittance of incident light [20].

When all three GWLs are present and the top GMR mode (in GWL_1_) couples with two other GMR modes (the middle GMR mode in GWL_2_ and the bottom GMR mode in GWL_3_), two sharp EIT-like characteristic resonances will appear simultaneously in the transmission dip that is caused by guided-mode resonance. The optical phenomenon resulting from the coupling of different GMR modes is the EIT-like spectral response, where two narrow transparent windows appear simultaneously in the stop band due to destructive interference [14,20,29]. The two EITs (denoted as EIT_1_ and EIT_2_ in Figure 1c) had respective resonance wavelengths *λ*_1_ and *λ*_2_ of 1549.95 nm and 1550.75 nm, and their transmittances reached 70.24% and 72.84%. The full widths at half maximum (*FWHM*) of the two EITs were 0.17 nm and 0.20 nm, respectively. Therefore, the *Q* factors of the two EITs were 9117 and 7753, respectively. Compared with the previous work that accidentally observed EIT-like characteristic spectra in several resonant grating structures [24,25], it is clear that we successfully used the GMR effect to achieve a double EIT-like effect with a narrow band and a high *Q* value in the resonance grating cascade structure.

In order to explore the mechanism of the observed double EIT-like effect, the electric field distribution of the double EIT peaks is shown in Figure 2a. It can be distinctly noticed that for the the EIT_1_ electric field, extremely strong coherent resonance appears in both GWL_1_ and GWL_2_, while for EIT_2_, strong coherent resonance mainly occurs in GWL_1_ and GWL_3_. At the resonance wavelength of EIT_1_ (EIT_2_), light is reflected back and forth between GWL_1_ and GWL_2_ (GWL_3_), and electromagnetic energy is coupled into GWLs with strong coherent resonance excited by modes coupling. The top GMR mode is coupled with the other two modes when Fabry–Pérot (F–P) resonance is introduced once the optical distance between grating G_1_ and grating G_2_ (G_3_) satisfies the phase matching condition [20,25,30], as shown in the following equation:(1)$$\varphi \;=\;\frac{2\pi \cdot n_{eff}\cdot S}{\lambda }=m\pi$$

Here, φ represents the phase accumulation from grating G_1_ to grating G_2_ (G_3_), *n_eff_* represents the effective refractive index from grating G_1_ to grating G_2_ (G_3_), S represents the optical distance from grating G_1_ to grating G_2_ (G_3_), and *λ* is the response wavelength of EIT, *m*∈N.

In other words, the modes in different GWLs are coupled once the optical system introduces F–P resonance. Thus, the GMR system shows two transparent peaks at 1549.95 nm and 1550.75 nm in the transmission spectrum.

The quasi-Λ four-level atomic energy system model is depicted in Figure 2b. The analogy between the GMR system and the atomic EIT system can make it easier to comprehend the physics of the double EIT-like effect. With |1 > being the ground state in the model, the fields of the bottom GMR mode and middle GMR mode correspond to the probability amplitude of the atom at the metastable state |2 > and |3 >, while the field of the top GMR mode is equivalent to the probability amplitude of the atom at the excited state |4 >. In addition, the probe field is the input of the top GMR mode, and the control fields are the coupling of the top GMR mode and the other two GMR modes. We observed that two EIT-like peaks appeared at the position of the probe mode after the control field was introduced [31,32]. EIT_1_ corresponds to the destructive interference between direct transition pathway |1 >→|4 > and indirect transition pathway |1 >→|4 >→|3 >→|4 >, and EIT_2_ corresponds to the interference between transition pathways |1 >→|4 > and |1 >→|4 >→|2 >→|4 >.

A narrow transparent window is usually associated with steep dispersion, which leads to a lower group velocity of light propagation; that is, slow light is a typical feature of the EIT effect. The spectral distribution of two EITs shows that there is extreme dispersion change near the two transparent windows. In the normal dispersion zone, the group velocity can be slower than *c* (the speed of light in vacuum), and slow light can be obtained [33]. There will also be fast light in the anomalous dispersion area, but we only pay attention to the slow light effect here. The performance of slow light can be determined by the optical delay time *τ* and the group index *n_g_*, which are defined as follows [34]:(2)$$\tau =\frac{d\varphi (\omega )}{d\omega }$$
(3)$$n_{g}=\frac{c}{v_{g}}=\frac{c}{L}\tau$$

In Equations (2) and (3), *φ*(*ω*) represents the transmission phase shift from the incident light to the phase monitor (placed at the bottom of GWL_3_), *c* is the speed of light, *n_g_* is the group velocity in the GMR system, and *L* is the length of the GMR system (from the top of the SiO_2_ coating to the bottom of GWL_3_). Using optical delay time to evaluate slow light performance is more precise as a result of different devices having different lengths. Figure 3a,b illustrate the phase shift and optical delay time of the double EIT-like effect shown in Figure 1. The two oblique phase shifts correspond to the two transparent windows. A positive delay time represents slow light, while a negative one is fast light. Two delay times of 22.59 ps and 8.43 ps are respectively generated near the two EIT peaks. Compared with other research using grating and the GMR effect to achieve slow light [35,36,37], our work had a quite excellent optical delay time, as shown in Table 1. Moreover, we obtained dual-channel slow light.

## 4. Discussion

Our previous research using the GMR effect to achieve the EIT-like effect in WGS [20,21] has discussed extensively and in depth the influence of parameters such as the width and period of the grating and the spacing between two GWLs on the transparent window. At the same time, while the response wavelengths of the GMR modes in different GWLs in the previous works were the same, the effect of the wavelength detuning between two different GMR modes on the transparent window has not been discussed. Here, we adjusted the wavelength detuning of the two GMR modes by changing the width of the grating to investigate the double EIT-like effect.

The transmission spectra and delay time at different *W*_2_ (different *δ*_1_) are given in Figure 4. We define the wavelength detuning of the top GMR mode and the middle GMR mode as *δ*_1_, where *δ*_1_
*=**λ_GMR_*_1_
*-λ_GMR_*_2_. In Figure 4a–d, *W*_2_ decreases from 380 nm to 335 nm, the detuning amount *δ*_1_ increases from 0.93 nm to 3.25 nm (*λ_GMR_*_1_ remains unchanged, *λ_GMR_*_2_ gradually decreases, and *λ_GMR_*_3_ is the same as *λ_GMR_*_1_ but not shown in Figure 4), EIT_1_ is blue-shifted from 1549.34 nm to 1547.45 nm, while the position of EIT_2_ remains unchanged. In order to show this more clearly, we present in Figure 4e the resonance wavelengths λ_1_ and λ_2_ of the two transparent windows under different *W*_2_ (different *δ*_1_). To study the effect of the detuning amount *δ*_1_ on the slow light performance, the corresponding delay time spectrum is given in Figure 4f. The delay time *τ*_2_ is kept around 10 ps, but the delay time *τ*_1_ decreases from 16.94 ps to 5.03 ps, since as the detuning *δ*_1_ increases, the coupling between the top GMR mode and the middle GMR mode becomes weaker.

Figure 5 demonstrates the transmission spectra and delay time under different *W*_3_ (different *δ*_2_). We define the wavelength detuning of the top GMR mode and the bottom GMR mode as *δ*_2_, where *δ*_2_
*=**λ_GMR_*_3_
*-λ_GMR_*_1_. It can be clearly seen that *W*_3_ increases from 415 nm (in Figure 5a) to 460 nm (in Figure 5d), and *δ*_2_ increases from 0.35 nm to 2.88 nm (*λ_GMR_*_1_ remains unchanged, *λ_GMR_*_3_ gradually increases, and *λ_GMR_*_2_ is equal to *λ_GMR_*_1_ but not shown in Figure 5). The position of EIT_1_ remains basically unchanged, while EIT_2_ is red-shifted from 1550.91 nm to 1551.97 nm. The resonance wavelengths of the two EITs at different *W*_3_ (different *δ*_2_) in Figure 5e show this more clearly. In order to explore the relationship between the detuning amount *δ*_2_ and the slow light performance, the delay time spectrum under different *W*_3_ (different *δ*_2_) is depicted in Figure 5f. Obviously, the delay time *τ*_1_ remains at ~30 ps, while as *W*_3_ increases, *δ*_2_ gradually increases. The delay time *τ*_2_ decreases significantly because the coupling between the top GMR mode and the bottom GMR mode becomes weaker.

The previous parameter discussion showed that the response wavelength detuning of different GMR modes will affect the position of the transparent window. We reduced the width *W*_3_ of the grating G_3_ to make the EIT_2_ blue shift close to EIT_1_, so that we obtained a wide-band flat-top transparent window (as shown in Figure 6a). The wide-band flat-top transparent window had a top bandpass width of about 0.72 nm and a peak transmittance of about 57%. The corresponding delay time spectrum is illustrated in Figure 6b. It can be seen that the delay time at the position of the transparent window is all above 2.4 ps. This not only provides an idea for the design of slow optical devices with a wide-band flat-top window but also can be used as a wide-band filter, where flat-top features can be used to separate each communication channel in optical communications applications.

## 5. Conclusions

In conclusion, we investigated the double EIT-like effect in the GMR system with three resonance grating cascade structures. This work provides a way to accomplish the double EIT-like effect by utilizing the GMR effect. The different GMR modes in the GWLs begin to couple once an F–P resonator is introduced, and two narrow-band transparent windows appear simultaneously in a transmission dip. We not only obtained quite excellent dual-channel slow light but also examined the effect of the wavelength detuning between different GMR modes on the position of the transparent window by changing the grating width. Meanwhile, a wide-band flat-top transparent window was also achieved by appropriately adjusting the wavelength detuning between different GMR modes. This work provides a method to achieve the double EIT-like effect, which not only has potential applications in dual-channel slow light devices but also in the field of optical communications.

## Figures and Tables

**Figure 1 materials-13-03710-f001:**
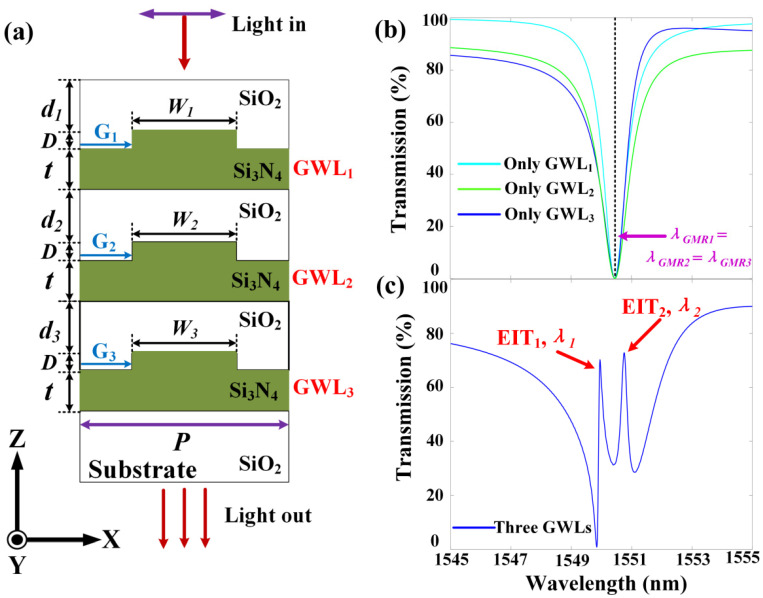
(**a**) Schematic diagram of the guided-mode resonance (GMR) system with three grating waveguide layers (GWLs); the geometric parameters are *d*_1_ = 630 nm, *d*_2_ = 690 nm, *d*_3_ = 690 nm, *D* = 200 nm, *t* = 900 nm, *W*_1_ = *W*_2_ = *W*_3_ = 400 nm, period *P* = 809 nm; (**b**) Transmission spectra of only GWL_1_ or only GWL_2_ or only GWL_3_; (**c**) Transmission spectrum when all three GWLs are present.

**Figure 2 materials-13-03710-f002:**
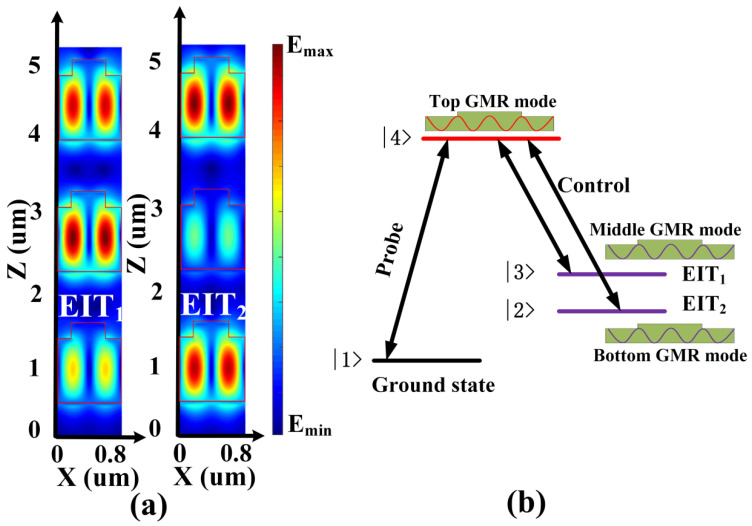
(**a**) Electric field distribution of two electromagnetically induced transparency (EIT) effects in the GMR system; (**b**) The analogy between the GMR system with a double EIT-like effect and a quasi-Λ four-level atomic system model.

**Figure 3 materials-13-03710-f003:**
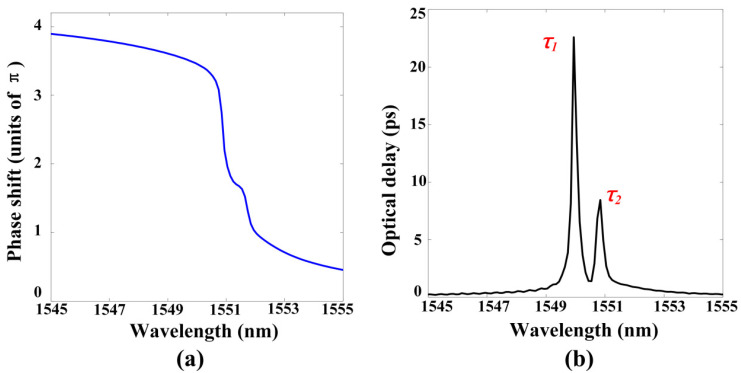
(**a**) The spectrum of phase shift in the GMR system with the double EIT-like effect in Figure 1; (**b**) The spectrum of delay time in the GMR system with the double EIT-like effect in Figure 1.

**Figure 4 materials-13-03710-f004:**
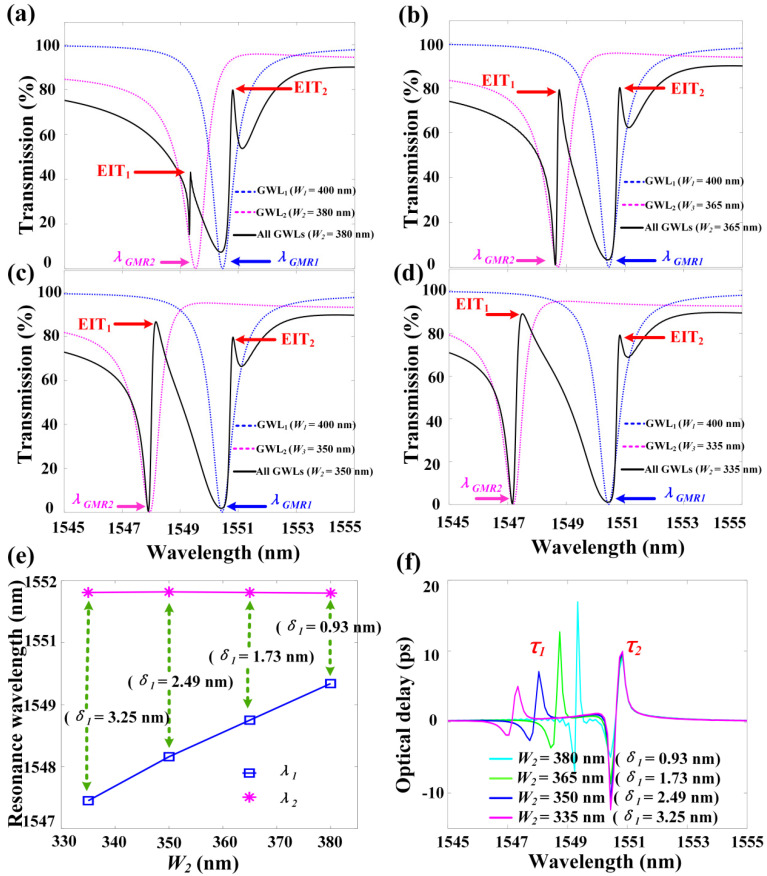
The transmission spectra and delay time under different *W*_2_ (different *δ*_1_); other parameters are the same as in Figure 1. (**a**) The transmission spectrum of the system when *W*_2_ = 380 nm (*δ*_1_ = 0.93 nm); (**b**) The transmission spectrum of the system when *W*_2_ = 365 nm (*δ*_1_ = 1.73 nm); (**c**) The transmission spectrum of the system when *W*_2_ = 350 nm (*δ*_1_ = 2.49 nm); (**d**) The transmission spectrum of the system when *W*_2_ = 335 nm (*δ*_1_ = 3.25 nm); (**e**) Two resonant wavelengths of two EITs under different *W*_2_ (different *δ*_1_); (**f**) Two delay times under different *W*_2_ (different *δ*_1_).

**Figure 5 materials-13-03710-f005:**
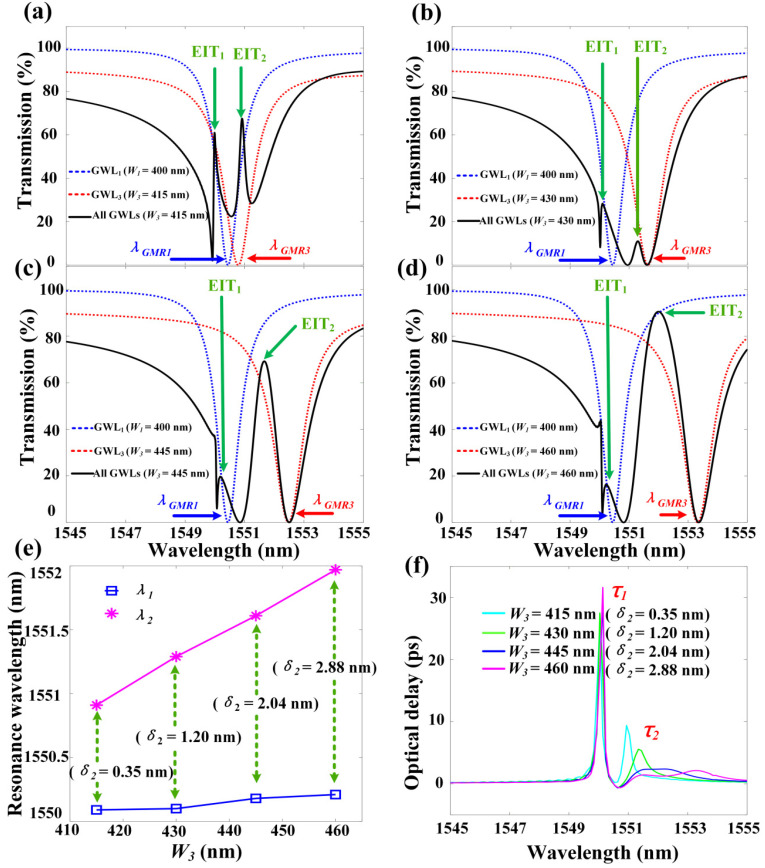
The transmission spectra and delay time under different *W*_3_ (different *δ*_2_); other parameters are the same as in Figure 1. (**a**) The transmission spectrum of the system when *W*_3_ = 415 nm (*δ*_2_ = 0.35 nm); (**b**) The transmission spectrum of the system when *W*_3_ = 430 nm (*δ*_2_ = 1.20 nm); (**c**) The transmission spectrum of the system when *W*_3_ = 445 nm (*δ*_2_ = 2.04 nm); (**d**) The transmission spectrum of the system when *W*_3_ = 460 nm (*δ*_2_ = 2.88 nm); (**e**) Two resonant wavelengths of two EITs under different *W*_3_ (different *δ*_2_); (**f**) Two delay times under different *W*_3_ (different *δ*_2_).

**Figure 6 materials-13-03710-f006:**
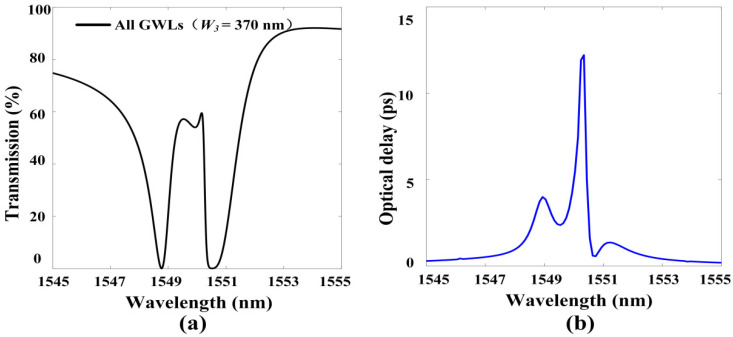
(**a**) Transmission spectrum of a wide-band flat-top transparent window, where the structural parameters are *W*_1_ = *W*_2_ = 400 nm, *W*_3_ = 370 nm; other parameters are the same as those in Figure 1; (**b**) The delay time spectrum corresponding to the wide-band flat-top transparent window.

**Table 1 materials-13-03710-t001:** Delay time compared to other slow light works based on grating and the GMR effect.

Reference	*τ*_1_ (ps)	*τ*_2_ (ps)
this work	22.59	8.43
35	29.73	–
36	21.80	–
37	3.10	–

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
