# Peer review of "Double Electromagnetically Induced Transparency and Its Slow Light Application Based on a Guided-Mode Resonance Grating Cascade Structure"

_materials, 2020, doi:10.3390/ma13173710_

Round 1
Reviewer 1 Report
The manuscript under review is dedicated to the numerical investigation of "double electromagnetically induced transparency" effect in cascaded guided-mode resonant gratings. The results are interesting and seem to be correct. In my opinion, the manuscript can be accepted for publication in Materials after a revision according to the following comments:
1. The investigation of cascaded resonant gratings and other cascaded resonant photonic structures is a research topic that has attracted considerable attention in recent years. Therefore, it is worth expanding the reference list and discuss in the introduction e.g. some of the following recent papers:
[1] Yeong Hwan Ko and Robert Magnusson, "Flat-top bandpass filters enabled by cascaded resonant gratings," Opt. Lett. 41, 4704-4707 (2016).
[2] Katsuaki Yamada, Kyu Jin Lee, Yeong Hwan Ko, Junichi Inoue, Kenji Kintaka, Shogo Ura, and Robert Magnusson, "Flat-top narrowband filters enabled by guided-mode resonance in two-level waveguides," Opt. Lett. 42, 4127-4130 (2017)
[3] Leonid L. Doskolovich, Evgeni A. Bezus, Dmitry A. Bykov, Nikita V. Golovastikov, and Victor A. Soifer, "Resonant properties of composite structures consisting of several resonant diffraction gratings," Opt. Express 27, 25814-25828 (2019).
[4] Leonid L. Doskolovich, Evgeni A. Bezus, and Dmitry A. Bykov, "Integrated flat-top reflection filters operating near bound states in the continuum," Photon. Res. 7, 1314-1322 (2019).
[5] Junxue Chen and Peixin Chu, "Phase-induced Fano antiresonance in a planar waveguide with two dielectric ridges," J. Opt. Soc. Am. B 36, 3417-3427 (2019).
In particular, in [3, 4], multiple resonant transmittance peaks similar to the presented "double-EIT" effect are obtained. Thus, is is worth comparing the results obtained in the paper under review with these works.
2. It seems that the formatting of the references in the text is wrong: the reference numbers are neither enclosed in square brackets, nor separated by commas or a dash (in the most cases).
3. From the caption to Fig. 1, it follows that all three gratings constituting the cascaded structure are identical. Why are the "single-grating" transmittance spectra for these three gratings shown in Fig. 1(b) different? This needs clarification.
4. At line 96, he authors mention the phase matching condition. They should describe it in more detail (probably, by providing a theoretical discussion with the corresponding equations). In addition, how were the used d2 and d3 values obtained?
5. Following the previous comment: how did the authors choose all the geometrical parameters of the initial structure discussed in Section 3? Were they optimized?
Reviewer 2 Report
The authors investigate a triple-layer guided mode resonance structure made
of silicon nitride on silicon substrates, performing 3D FDTD simulations.
They identify a fully optical analog of double electromagnetically induced
transparency, present its atomic level structure analog, and discuss the
properties of the double EIT, such as the transmission (absorption) and
phase shift (dispersion) spectra and optical delay time (inverse group
velocity), in dependence on the parameters of the GMR system. In particular,
the authors identify a regime when the two EIT peaks approach each other to
form a wide-band flat-top transparency window for the probe light.
The result reported in this paper seem to be valid, interesting and
potentially important for various practical applications. The paper is
clear and well organized. Perhaps minor corrections of the English
language can be made by the copy-editors.
I therefore recommend this paper for publication.
Reviewer 3 Report
This manuscript presents numerical calculations of light transmission through a system with three grating layers of Si_3N_4. In transmission spectra there are two sharp peaks which the authors call as electromagnetically induced transparency. The manuscript is extremely similar to Ref [20]. Actually this is practically the same work only difference is that in Ref. [20] it was considered not three but two grating layers. I can assume that the authors will soon consider 4 layers, then five and so on. Authors should explain why such a simple multiplication of the same result should be printed in a serious scientific journal.
In my opinion, the observed phenomenon has a simpler explanation. Each layer separately gives a resonance minimum at the same frequency, since the layers are practically the same [see Fig. 1 (b)]. When three layers are taken together, the resonant frequency of each layer shifts slightly due to the interaction between the layers. Figure 1(c) just can be interpreted as three minima but not as two maxima. But that is just my guess, maybe I am not correct.
Technical comments:
1) In the third line of the Introduction, it is given reference 12 instead of References 1-2. The same happens in many other places.
2) Abbreviation GMR should be explained in the main text of the manuscript, not only in Abstract.
3) Reference PRB 77, 205113 (2008) probably should be also mentioned.
Reviewer 4 Report
The authors report a theoretical study on a double electromagnetically induced transparency and its slow light application based on guided mode resonance grating cascade strcuture. The paper is well organized and the aim of the study is clearly presented. This is a quite interesting paper and the topic of the manuscript is relevant to the field of the journal. However, it can not be published as it is. Minor rvesions are necessary.
The following revisions of the manuscript are required in order to make it easier to understand:
Scientific Comments:
- What is the influence of the period p of the grating on the results obtained ?
- For, the study of the delay time as of the function of the wi, the authors should precise that the different waveguides are well single-mode.
- The delay time was studied as a function of the wi. Which is the influence of the di on the delay time?
Round 2
Reviewer 3 Report
I have already reviewed this manuscript. The authors took into account some of my comments and I can recommend the manuscript for publication.